# Maximizing Bio-Hydrogen Production from an Innovative Microbial Electrolysis Cell Using Artificial Intelligence

Ahmed Fathy [1,*], Hegazy Rezk [2,3], Dalia Yousri [4], Abdullah G. Alharbi [1], Sulaiman Alshammari [1] and Yahia B. Hassan [5]

1   Electrical Engineering Department, Faculty of Engineering, Jouf University, Sakaka 72388, Saudi Arabia
2   Department of Electrical Engineering, College of Engineering in Wadi Alddawasir, Prince Sattam bin Abdulaziz University, Wadi Alddawasir 11991, Saudi Arabia
3   Electrical Engineering Department, Faculty of Engineering, Minia University, Minia 61111, Egypt
4   Electrical Engineering Department, Faculty of Engineering, Fayoum University, Fayoum 63514, Egypt
5   Electrical Engineering Department, Higher Institute of Engineering, Minia 61519, Egypt
*   Correspondence: afali@ju.edu.sa

**Abstract:** In this research work, the best operating conditions of microbial electrolysis cells (MECs) were identified using artificial intelligence and modern optimization. MECs are innovative materials that can be used for simultaneous wastewater treatment and bio-hydrogen production. The main objective is the maximization of bio-hydrogen production during the wastewater treatment process by MECs. The suggested strategy contains two main stages: modelling and optimal parameter identification. Firstly, using adaptive neuro-Fuzzy inference system (ANFIS) modelling, an accurate model of the MES was created. Secondly, the optimal parameters of the operating conditions were determined using the jellyfish optimizer (JO). Three operating variables were studied: incubation temperature (°C), initial potential of hydrogen (pH), and influent chemical oxygen demand (COD) concentration (%). Using some measured data points, the ANFIS model was built for simulating the output of MFC considering the operating parameters. Afterward, a jellyfish optimizer was applied to determine the optimal temperature, initial pH, and influent COD concentration values. To demonstrate the accuracy of the proposed strategy, a comparison with previous approaches was conducted. For the modelling stage, compared with the response surface methodology (RSM), the coefficient of determination increased from 0.8953 using RSM to 0.963 using ANFIS, by around 7.56%. In addition, the RMSE decreased from 0.1924 (using RSM) to 0.0302 using ANFIS, whereas for the optimal parameter identification stage, the optimal values were 30.2 °C, 6.53, and 59.98 (%), respectively, for the incubation temperature, the initial potential of hydrogen (pH), and the influent COD concentration. Under this condition, the maximum rate of the hydrogen production is 1.252 $m^3H_2/m^3d$. Therefore, the proposed strategy successfully increased the hydrogen production from 1.1747 $m^3H_2/m^3d$ to 1.253 $m^3H_2/m^3d$ by around 6.7% compared to RSM.

**Keywords:** bio-hydrogen; microbial electrolysis cell; artificial intelligence; optimization

## 1. Introduction

High population growth generates massive waste and wastewater; furthermore, fossil fuel's extensive consumption leads to greenhouse gas emissions and fossil fuel exhaustion with time [1–3]. On the other hand, organic wastes, including wastewater, contain huge amounts of stored energy [4,5]. Thus, transforming organic wastes into energy or energy carriers is considered a quixotic solution [6,7]. Many approaches such as gasification are proposed to convert the wastes to chemical energy and energy carriers such as ethanol, methanol, and hydrogen [8–13]. Among different energy carriers, hydrogen is the cleanest and most promising one. Only water is produced, it has a high energy density, it can be used in many applications such as ammonia or methanol production, and can be used as a transportation fuel [14–16].

Hydrogen demand has recently utilizing due to the augmentation of the number of cars operated with hydrogen fuel cells [17,18]. Currently, fossil fuel is used to produce the most hydrogen (~50 million tons/year) through coal gasification and methane reforming, leading to immense greenhouse gas emissions [19,20]. Researchers have focused on finding novel approaches to utilizing organic wastewater for producing hydrogen, such as microbial electrolysis cells (MECs), fermentation, and gasification [4,21–23]. A microbial electrolysis cell (MEC) is a bio electrochemical cell that converts organic materials including wastes to hydrogen gas via the catalytic activity of living microorganisms [24]. The structure of an MEC is similar to that of the microbial fuel cell, consisting of an anode and cathode connected to a power supply with or without a separator. The electroactive biofilm in the anode side consumes the organic wastes in the wastewater, generating protons and electrons which are transferred to the cathode and combined to produce hydrogen. Compared to water electrolysis, an MEC requires less external voltage (0.3–0.9 V) than water electrolysis (~1.23 V) [25,26]. Additionally, the hydrogen recovery from MECs is much higher than dark fermentation (20%) [27]. Different wastewater could be used as substrates in applying MECs, such as industrial wastewater, domestic wastewater, and food processing wastewater [28–31]. A dual tubular chamber MEC was fed with domestic wastewater using 0.9 V (applied voltage), achieving a high coulombic efficiency of 98.5% with a hydrogen production rate of $0.18 \pm 0.03$ m$^3$/m$^3$.d and 151.9% energy recovery [32]. A single chamber MEC was assembled using an MoS$_2$-GO nickel foam cathode and operated with dye wastewater (alizarin yellow R) to produce hydrogen simultaneously with dye removal [33]. Different parameters including dye concentration, buffer, and co-substrates were investigated. A 90% dye removal was achieved within 10 h using an initial dye concentration of 30–150 mg/L and sodium acetate and phosphate buffer, indicating that the alizarin yellow R dye did not inhibit the electrochemically active microorganism. The highest hydrogen production rate of 0.183 m$^3$/m$^3$.d and chemical oxygen demand (COD) removal of 92.44% were achieved with phosphate buffer and co-substrate (sodium acetate and glucose).

The palm oil Industry produces a huge amount of palm oil mill effluent (POME) annually (48–72 million metric tons) [34]. Treatment of POME before charging to the environment is essential. POME can be used as a fuel in MECs to produce hydrogen. The maximum benefit of the MEC can be achieved through operating it under optimum conditions.

Hydrogen production via photosynthetic bacteria from sole and compound carbon sources have been examined by Han et al. [35]. The results confirmed that the hydrogen production for compound carbon sources was better than sole carbon sources.

Without having an accurate model for the process under investigation, it is not possible to rely on its results. Both Fuzzy logic (FL) and artificial neural networks (ANNs) have their own merits. FL can effectively handle input–output data of a system with nonlinear relationships [36]. However, ANN can be trained with a training algorithm such as backpropagation (BP) [37,38]. Certainly, combining both techniques together will definitely produce a more powerful tool. ANFIS merges the concepts of both Fuzzy logic and neural networks in one arrangement. This inspired the authors to apply the ANFIS in building the model of the microbial electrolysis cell (MEC). The ANFIS simulates bio-hydrogen production during the wastewater treatment process by MECs in terms of incubation temperature (°C), initial potential of hydrogen (pH), and influent COD concentration (%). The contributions of this work can be outlined as follows:

- A reliable ANFIS model is proposed to simulate the microbial electrolysis cell.
- The optimal values of incubation temperature, initial pH, and influent COD concentration are determined using a jellyfish optimizer.
- The suppository of the proposed methodology is proved.
- The production of bio-hydrogen capacity is maximized.

The reminder of the paper is arranged as follows. The data that were used for constructing the model of the microbial electrolysis cell are presented in Section 2. The two

main phases of the proposed strategy, modelling and parameter estimation, are explained in Section 3. The results of the modelling and optimization are discussed in Section 4. Finally, the main findings and future work are outlined in Section 5.

## 2. Dataset

The dataset (License Number-5437491197207) [39] in this paper was used with permission. A single chamber membraneless MEC consisting of DURAN glass bottle of 800 mL with 500 mL working volume was used to produce hydrogen from POME. The composition of the produced gas ($H_2$, $CH_4$, and $N_2$) was analyzed using gas chromatography (GC), model SRI 8600C, SRI Instruments, USA, armed with helium (He) ionization and a thermal conductivity detector (TCD). A cleaned graphite plate, an isomolded cuboid of 100 cm, and nickel electro-formed meshing of 84 $cm^2$ were used as anode and cathode, respectively. The distance between the two electrodes was kept at 2.5 cm, and they were connected with titanium wire across 10 Ω external resistance. A POME anaerobic sludge from the anaerobic pond (palm oil mill factory in Kuala Lumpur, Malaysia) was used as inoculum and POME wastewater of 482 mg/L at pH of 4.6 was filtrated and sterilized. Then, the wastewater was diluted with sodium phosphate buffer (50 mM, pH 7) and 400 mL of pretreated POME was injected into the MEC and used as the fuel. An amount of 100 mL of enriched inoculum of POME anaerobic sludge was inoculated in the MEC.

The anode biofilm was enriched for more than one month in the fed-batch mode of microbial fuel cells to promote the electrogenic microorganism on the anode surface. The current/voltage generation and COD were recorded to determine the performance of the MEC. The produced gas was collected in the upper part of the bottle (300 mL). The gas wass compiled using a graduated measuring cylinder of 2000 mL filled with deionized water and attached with an inverted Viton tubing. After each cycle, the MEC cell was subjected to dry air and atmospheric air for 30–45 min to prevent methane production by inhibiting methanogenic microorganisms. The MEC was filled with fresh substrate and the inoculum. Then, the cell was purged with nitrogen for 20 min. For more information, refer to [39].

## 3. The Proposed Strategy

In this work, the anticipated strategy includes two phases as explained in Figure 1: modelling using ANFIS and optimal parameter estimation using JO.

### 3.1. Model of MEC

The model of the MEC was created using an adaptive neuro-Fuzzy inference system (ANFIS). ANFIS is a form of artificial neural network that is based on the Takagi–Sugeno fuzzy inference system. As ANFIS integrates both neural networks and Fuzzy logic, it has the potential to capture the benefits of both in a single structure. During the fuzzification phase, the degree of the input parameters is defined by membership functions (MFs), whereas through the inference phase, the Fuzzy rules will be created. In the last phase, defuzzification, the output is translated into crisp [40]. Despite there being different MF forms and defuzzification methods, the gaussian shape and weight average are adopted.

In data modelling, the Sugeno-type IF-THEN Fuzzy rule is the most suitable one, as its consequential part is a mathematical function of the inputs. Hence, it can efficiently capture the functional relationship (dependency) between the system's output and its inputs through a training algorithm. It is worth mentioning that the other common type of the IF-THEN Fuzzy rule is the Mamdani-type, which is more suitable in dealing with control system applications where the rules are usually configured using an expert. An example of the ANFIS IF-THEN rule statements can be represented as follows:

$$\text{IF } a \text{ is } x_1 \text{ and } b \text{ is } y_1 \text{ then } f_1 = g_1(a, b) \tag{1}$$

$$\text{IF } a \text{ is } x_2 \text{ and } b \text{ is } y_2 \text{ then } f_2 = g_2(a, b) \tag{2}$$

where $x$ and $y$ are the MFs of input parameters $a$ and $b$.

$$f = \widetilde{\omega}_1 f_1 + \widetilde{\omega}_2 f_2 \qquad (\text{Output Layer}) \qquad (3)$$

Evaluating $\widetilde{\omega}_1 g_1(x, y)$ and $\widetilde{\omega}_2 g_2(x, y)$ (defuzzification layer) is achieved using the following:

$$\widetilde{\omega}_1 = \frac{\omega_1}{\omega_1 + \omega_2} \ and \ \widetilde{\omega}_2 = \frac{\omega_2}{\omega_1 + \omega_2} \qquad (\text{N Layer}) \qquad (4)$$

$$\omega_1 = \mu_{A_1} * \mu_{B_1} \ and \ \omega_2 = \mu_{A_2} * \mu_{B_2} \qquad (\pi \text{ Layer}) \qquad (5)$$

where $\mu_{A_1}$, $\mu_{A_2}$, $\mu_{B_1}$ and $\mu_{B_2}$ are the MF values of the two inputs.

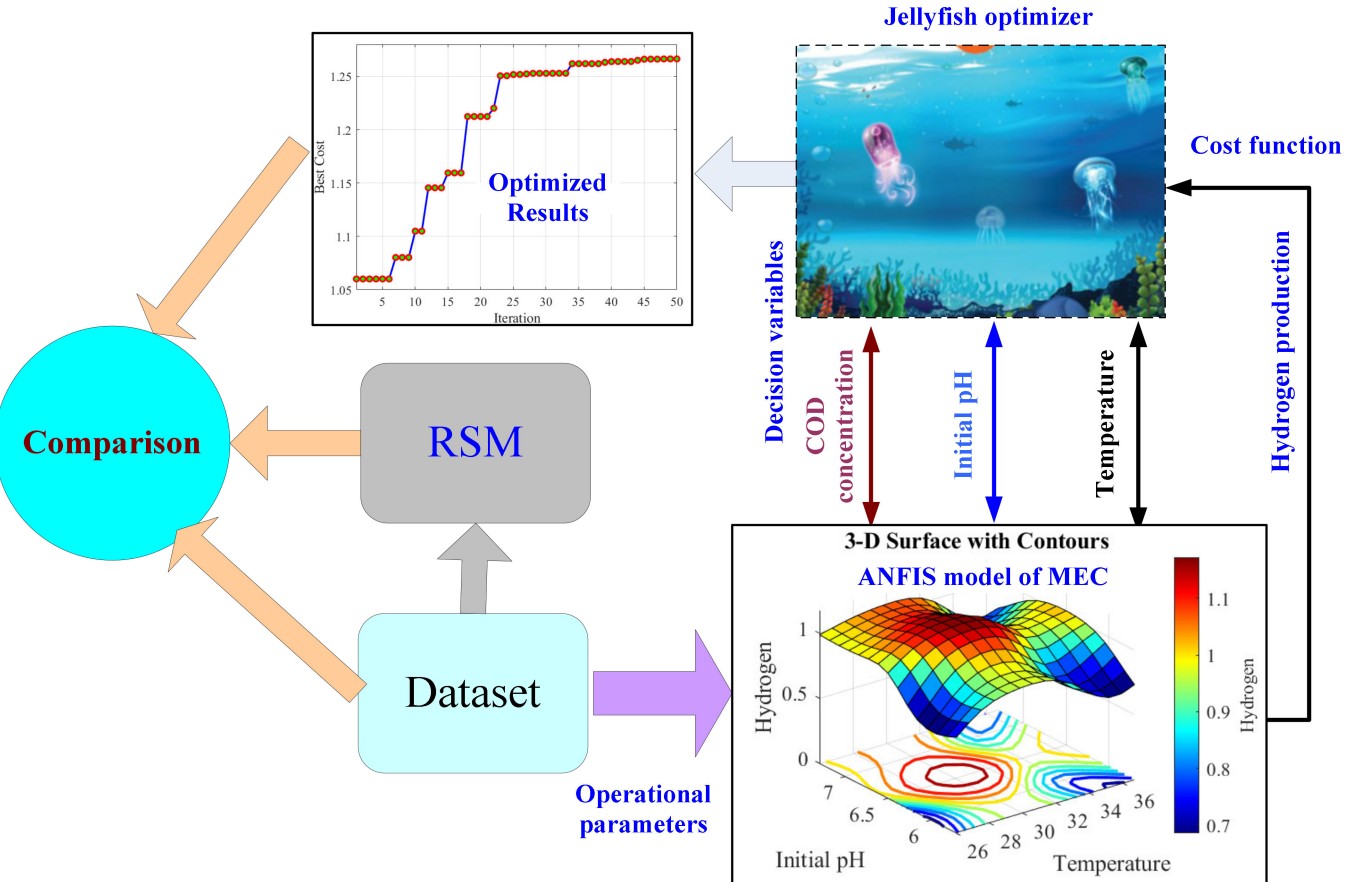

**Figure 1.** Layout of the proposed methodology.

### 3.2. Parameter Estimation Using Jellyfish Optimizer

The aim of the parameter estimation process is determining the best parameters of incubation temperature, initial pH, and influent COD concentration to maximize the bio-hydrogen production during the wastewater treatment process by the MEC. Therefore, after obtaining a robust ANFIS model of the MEC, the jellyfish optimizer is utilized to determine the optimum parameters of controlling inputs.

The optimization statement can be formulated by the following:

$$x = \underset{x \in R}{\operatorname{argmax}}(y) \qquad (6)$$

where $x$ is the controlling inputs and $y$ is bio-hydrogen production.

The jellyfish optimizer (JO) is employed to perform the parameter estimation process.

Jellyfish live in water of different depths and temperatures around the world. They are shaped like bells; some have a diameter of less than a centimeter while others are

very large. They have a wide range of colors, sizes, and shapes. The two major stages of metaheuristics are exploration and exploitation. In the JSO, movement toward an ocean current is exploration, movement within a jellyfish swarm is exploitation, and there is a time control [41]. A time control method controls the changing between various forms of movement.

**Ocean current**: The ocean current has large quantities of nutrients; therefore, the jellyfish are attracted to it. The current ocean direction $(\overrightarrow{co})$ is defined by averaging all the vectors from each jellyfish in the ocean to jellyfish that are currently in the best location [41].

$$d^r = \frac{1}{n}\sum d_i^r = \frac{1}{n}\sum(X_{best} - e_c X_i) = X_{best} - e_c \times \left(\frac{\sum X_i}{n}\right) = X_{best} - e_c\mu = X_{best} - df \quad (7)$$

where $n$ is the population size, $X_{best}$ is the best achieved location, $e_c$ is a convergence factor, $\mu$ is the mean of the jellyfish positions, and $df$ is the difference between the best location and the mean of the jellyfish locations.

The new position is defined by the following relation.

$$X_i(t+1) = X_i(t) + r(X_{best} - \beta \cdot r)\mu \quad (8)$$

where $r$ is a random number in range [0, 1] and $\beta$ denotes the positive distribution coefficient ($\beta = 3$).

**Jellyfish swarm**: In a swarm, jellyfish have passive (type A) and active (type B) motions, respectively. At first, when the swarm has just been formed, most jellyfish exhibit type A movement. Over time, they gradually exhibit type B movement [41]. Type A is defined using the following relation.

$$X_i(t+1) = X_i(t) + \gamma \cdot r(Ub - Lb) \quad (9)$$

where $Ub$ and $Lb$ are the maximum and minimum limits and $\gamma$ represents a movement factor proportional to the length of movement around the jellyfish's position. In this work, the value of $\gamma$ is 0.1.

Type B movement is defined using the following relation:

$$X_i(t+1) = X_i(t) + r \cdot D^r$$

$$D^r = \begin{cases} X_j(t) - X_i(t) & if \ fit(X_i) \geq fit(X_j) \\ X_i(t) - X_j(t) & if \ fit(X_i) < fit(X_j) \end{cases} \quad (10)$$

where $fit$ denotes the cost function and $j$ is selected arbitrarily.

The time control mechanism is as follows.

The ocean current includes large quantities of nutritious food, so jellyfish are attracted by it. Over time, extra jellyfish gather together, and a swarm is established. As the temperature or wind varies the ocean current, the jellyfish in the swarm move toward another ocean current and another jellyfish swarm is established. The movements of jellyfish inside a jellyfish swarm are type A (passive motions) and type B (active motions), between which the jellyfish switch. Type A is preferred in the beginning; as time goes by, type B is desired. The time control mechanism, $c(t)$, switches the movement type between jellyfish swarm and ocean current. The time control can be defined as follows.

$$c(t) = \left|\left(1 - \frac{t}{T_{max}}\right) \cdot (2r - 1)\right| \quad (11)$$

where $T_{max}$ represents the max number of iterations. In this work, $T_{max}$ is assigned to 50.

## 4. Results and Discussion

### 4.1. ANFIS Modelling

The bio-hydrogen production of a microbial electrolysis cell is simulated in terms of three parameters: the incubation temperature, the initial pH, and the influent COD concentration. To achieve a smoother prediction curve, the gaussian shape is a very suitable MF, as it provides a good transition from a predicted point to the next, contrary to the other triangular or trapezoidal shapes which produce jumps in the predictions. Consequently, in data-based modelling, the gaussian shape performs better, however, in control systems, triangular and trapezoidal shapes are the most suitable MFs. On the other hand, the SC technique produces the optimal as well as the smallest number of rules that can effectively handle the trend of the dataset. In the current research, the considered data are composed of 16 points. To train and test the ANFIS model, these points are separated into two sections with a ratio of 70:30. Therefore, 11 points are assigned for training and 5 points are used for testing. The sub-cluster is used to create the ANFIS rules, of which there are eleven, and number of epochs is selected to two for avoiding overfitting. To construct a strong model, the procedure of continuous training was conducted for the ANFIS model up to an achieved stopping criterion. In the current work, the training procedure stops when the testing data mean squared error (MSE) obtains the lowest value. Therefore, the model is trained up to when a smaller MSE is achieved. The statistical indicators are estimated in both stages, the training and the testing, to evaluate the performance of the resulting model in every stage. Table 1 demonstrates the values of mean squared error (MSE), root mean squared error (RMSE), and coefficient of determination ($R^2$). The first two indicators (MSE and RMSE) determine the accuracy of the model's predictions, while the latter ($R^2$) determines the tracking ability. The model is said to be better when its predictions provide the lowest MSE and the largest $R^2$ values for both groups of datasets (training and testing). It is worth mentioning that $R^2$ is the squared value of the correlation coefficient, which implies that its value is in the range [0, 1], where 0 and 1 indicate no correlation and full correlation between the predicted and actual signals, respectively. The resultant statistical markers of the ANFIS model are presented in Table 1.

**Table 1.** MSE, RMSE, and *R*-squared values of the model.

| MSE | | | RMSE | | | *R*-Squared | | |
|---|---|---|---|---|---|---|---|---|
| Training | Testing | All | Training | Testing | All | Training | Testing | All |
| $2.74 \times 10^{-13}$ | 0.0029 | 0.0009 | $5.23 \times 10^{-7}$ | 0.0541 | 0.0302 | 1.0 | 0.973 | 0.963 |

Referring to Table 1, the MSE values are $2.74 \times 10^{-13}$ and $5.23 \times 10^{-7}$ during training and testing, respectively, whereas the RMSE values are 0.0029 and 0.0541 during training and testing, respectively. The coefficients of determination are 1.0 and 0.973, respectively, for training and testing. Compared with the response surface methodology (RSM [39]), the coefficient of determination increased from 0.8953 using RSM to 0.963 using ANFIS, by around 7.56%. In addition, thanks to ANFIS modelling, the RMSE decreased from 0.1924 (RSM) to 0.0302 using ANFIS. Lowering the RMSE and boosting the coefficient of determination prove the successful ANFIS modelling phase and can be used for the model predictions. Figure 2 shows three-input one-output configuration of the model. The three input parameters are the incubation temperature, the initial pH, and the influent COD concentration, and the output is hydrogen rate. In the current model, the subtractive clustering method produced a lower number of rules, which is 11, as demonstrated in Figure 3. However, the MFs are shown in Figure 3, and it is worth mentioning that every colored MF curve represents a cluster in the input space.

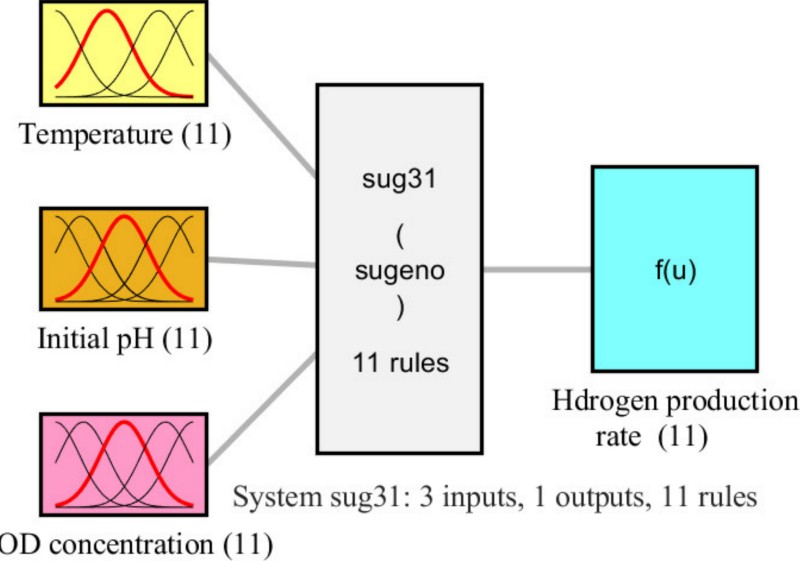

**Figure 2.** Structure of ANFIS model of microbial electrolysis cell.

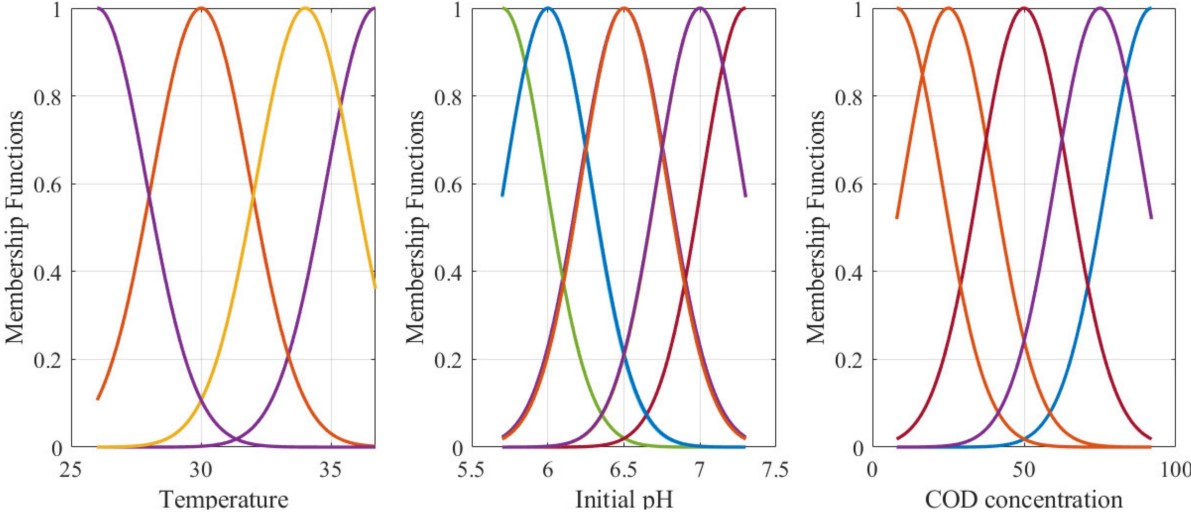

**Figure 3.** Membership functions of ANFIS of the microbial electrolysis cell.

Figure 4 illustrates the three-dimensional structure of the ANFIS model. The interaction between controlling factors, i.e., initial concentration of the COD, operating cell temperature, and pH on bio-hydrogen production, is shown. The highest values of the output go to the dark red, but the lowest values go to the dark blue. The figure shows that the interaction between every two factors positively affects bio-hydrogen productivity compared to the single factor. As is clear from the figure, the increase in the initial COD resulted in increasing the bio-hydrogen production up to 60%, and then the bio-hydrogen production is decreased with the further increase in the COD concentration. Similarly, it is noticed that the bio-hydrogen productivity increases with increasing the pH up to 6.5, then decreases with the further increase in the pH. In the same manner, the best bio-hydrogen production is achieved at an optimum operating cell temperature of 30 °C. The improved bio-hydrogen production by increasing the COD up to 60% would be related to the increase in the COD (fuel) used by the microorganism for bio-hydrogen production. On the other hand, at higher COD values, the bio-hydrogen decreased due to the poisoning effect of higher COD values on the microorganisms [39,42,43]. The best bio-hydrogen productivity obtained at pH 6.5 is related to the high microbial activity at moderate pH values compared to the acidic (lower pH values) or alkaline (higher pH values) [44–46]. Similarly, the activity

of the microorganisms increases with increasing the cell temperature up to around 30 °C. However, at higher cell temperatures the activity of the microorganisms decreases, and thus bio-hydrogen production decreases [47,48].

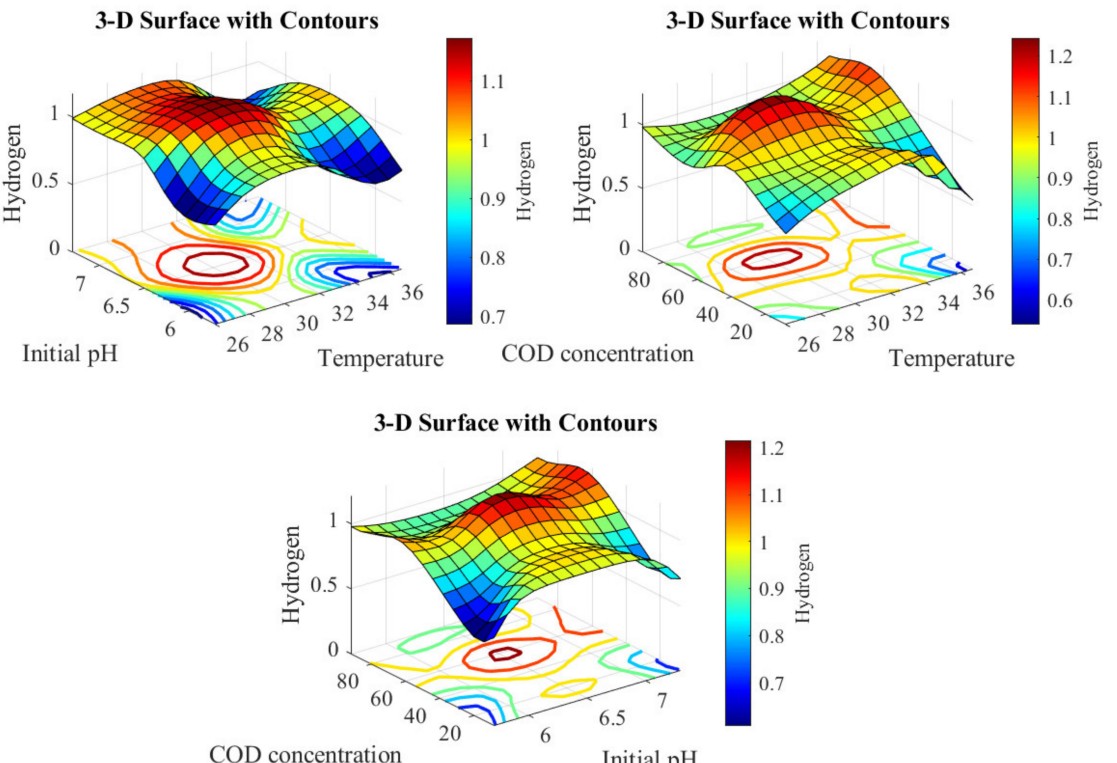

**Figure 4.** ANFIS surface of microbial electrolysis cell. Hydrogen production ($m^3 H_2/m^3 d$), temperature (°C), and COD concentration (%).

Capturing the correct relation between the produced bio-hydrogen and the three controlling variables, incubation temperature, initial pH, and influent COD concentration, encourages the ANFIS model to predict the produced bio-hydrogen accurately. The predicted versus measured data of the ANFIS model of the MEC are presented in Figure 5. It can be seen that there is good agreement between the experimental samples and those estimated by the ANFIS-based model.

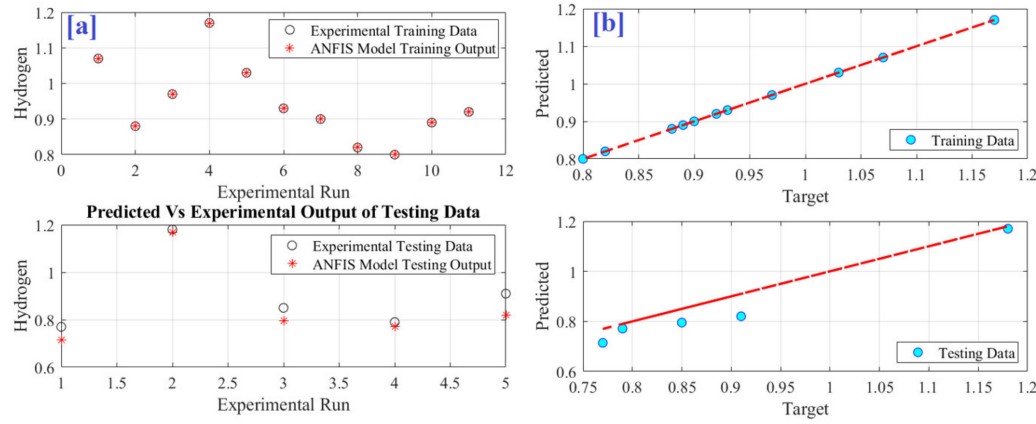

**Figure 5.** Predicted versus measured data of ANFIS model of microbial electrolysis cell (hydrogen production $m^3 H_2/m^3 d$). (**a**) predicted versus measured data and (**b**) prediction accuracy.

One more significant step after completing the ANFIS model is model verification. Table 2 gives the verification results compared with the measured samples. Experimentally, the hydrogen production is 1.1747 $m^3H_2/m^3d$ under the values 30.23 °C, 6.63, and 50.71%, respectively, for incubation temperature, initial pH, and influent concentration. Considering Table 2, the difference between the measured value and that predicted by ANFIS is 0.0018 $m^3H_2/m^3d$. It is 0.15% from the reference value, so it is considered as very small error.

**Table 2.** ANFIS model validation.

| Strategy | Incubation Temperature (°C) | Initial pH | COD Concentration (%) | Hydrogen Production $m^3H_2/m^3d$ |
|---|---|---|---|---|
| Measured [37] | 30.23 | 6.63 | 50.71 | 1.1747 |
| ANFIS model | 30.23 | 6.63 | 50.71 | 1.1765 |

*4.2. Parameter Identification Process*

To verify the reliability of JO, the optimization process is executed for 30 runs and some statistical analysis was performed. During the optimization process, the three input control parameters, the incubation temperature (°C), the initial potential of hydrogen (pH), and the influent COD concentration (%), were used as decision variables, whereas the objective function that was required to be a maximum is the hydrogen production rate. The number of particles and maximum number of iterations were 3 and 50, respectively. Figure 6 shows the obtained optimization results. The cost function values during 30 runs are presented in Figure 6a, and the minimum, maximum, average, and standard deviation are 1.1179, 1.253, 1.2332, and 0.03, respectively. It is obvious that the proposed JO succeeded in reaching the maximum capacity of hydrogen (1.253 $m^3H_2/m^3d$)>. The particle convergence of temperature, initial pH, and COD concentration are demonstrated in Figure 6c, Figure 6d, and Figure 6e, respectively. It can be seen that the optimal value of the incubation temperature is converged to 30.2 °C, and the best value of the initial potential of hydrogen (pH) is converged to 6.53. In addition, the best influent COD concentration percent is converged to 59.98 (%). Table 3 presents the optimized parameters obtained by the experiment by the response surface methodology (RSM) and by the proposed JO and ANFIS. Considering Table 3, the integration between JO and ANFIS confirmed the excellent performance in terms of improved hydrogen production rate. The predicted hydrogen production rate was increased from 1.18 $m^3H_2/m^3d$ by experimental work to 1.253 $m^3H_2/m^3d$ using the JO and increased from 1.1747 $m^3H_2/m^3d$ by response surface methodology to 1.253 $m^3H_2/m^3d$ using the JO. In sum, hydrogen production increased by around 6.2% and 6.7%, respectively, compared to experimental work and response surface methodology.

**Table 3.** Optimal parameters using different methods.

| Strategy | Incubation Temperature (°C) | Initial pH | COD Concentration (%) | Hydrogen Production $m^3H_2/m^3d$ |
|---|---|---|---|---|
| Measured [39] | 30.0 | 6.5 | 50 | 1.18 |
| RSM [39] | 30.23 | 6.63 | 50.71 | 1.1747 |
| JO and ANFIS (predicted) | 30.2 | 6.53 | 59.98 | 1.252 |

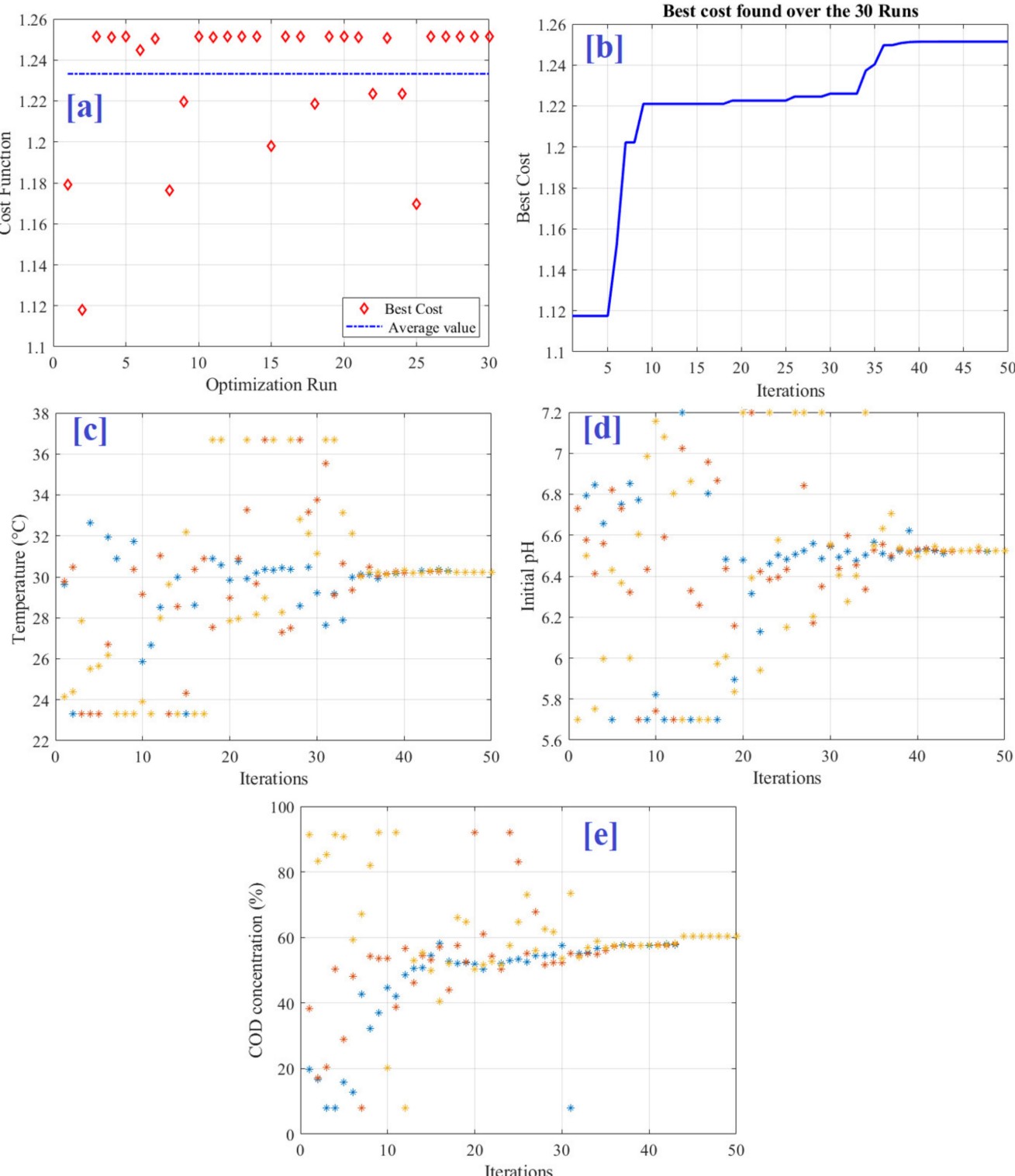

**Figure 6.** Optimization results: (**a**) cost function values of 30 runs, (**b**) best objective function variation (hydrogen production m$^3$H$_2$/m$^3$d), (**c**) temperature (°C), (**d**) initial pH, and (**e**) COD concentration (%).

## 5. Conclusions

Three essential operating variables, incubation temperature, initial pH, and influent COD concentration, are mainly influenced by the bio-hydrogen production during the wastewater treatment process by the microbial electrolysis cell (MEC). Therefore, estimating the best value of these variables represents a challenge. Two phases were implemented: AN-FIS modelling and the parameter identification process. Regarding the ANFIS modelling, a

robust model was created based on measured samples. The coefficient of determination increased from 0.8953 using RSM to 0.963 using ANFIS, by around 7.56%. As well, thanks to ANFIS modelling, the RMSE decreased from 0.1924 (RSM) to 0.0302. Lowering the RMSE and boosting the coefficient of determination prove the successful ANFIS modelling phase.

Regarding the model validation, experimentally, the hydrogen production is 1.1747 under the values 30.23 °C, 6.63, and 50.71%, respectively, for incubation temperature, initial pH, and influent COD concentration. The difference between the measured value and that predicted by ANFIS is 0.0018 $m^3H_2/m^3d$. This difference is very small and acceptable. Then, the jellyfish optimizer (JO) was used to identify the best parameters of input variables. The optimal values are 30.2 °C, 6.53, and 59.98 (%), respectively, for the incubation temperature, the initial potential of hydrogen (pH), and the influent COD concentration. Under this condition, the maximum rate of the hydrogen production is 1.252 $m^3H_2/m^3d$. Therefore, the JO and ANFIS confirmed the superior performance in boosting hydrogen production. The hydrogen production increased from 1.18 $m^3H_2/m^3d$ to 1.253 $m^3H_2/m^3d$ by around 6.2% compared with experimental work. In addition, it boosted from 1.1747 $m^3H_2/m^3d$ to 1.253 $m^3H_2/m^3d$, around 6.7% compared with RSM. Finally, in the future work the optimized results will be verified experimentally.

**Author Contributions:** Supervision, conceptualization, writing—review and editing, methodology, software, formal analysis, A.F. and H.R.; Methodology, software, validation, formal analysis, writing—review and editing, D.Y. and A.G.A.; Resources, data curation, investigation, formal analysis, methodology, software, validation, writing—original draft preparation, S.A. and Y.B.H. All authors have read and agreed to the published version of the manuscript.

**Funding:** This work was funded by the Deanship of Scientific Research at Jouf University under Grant Number (DSR2022-RG-0108).

**Institutional Review Board Statement:** Not applicable.

**Informed Consent Statement:** Not applicable.

**Data Availability Statement:** Not applicable.

**Conflicts of Interest:** The authors declare no conflict of interest.

## Abbreviations

| | |
|---|---|
| MEC | microbial electrolysis cell |
| COD | chemical oxygen demand |
| POME | palm oil mill effluent |
| JO | jellyfish optimizer |
| ANFIS | adaptive neuro-Fuzzy inference system |
| FL | Fuzzy logic |
| MFs | membership functions |

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
