# Peer review of "Maximizing Bio-Hydrogen Production from an Innovative Microbial Electrolysis Cell Using Artificial Intelligence"

_sustainability, doi:10.3390/su15043730_

Round 1

Reviewer 1 Report

The Abstract could be improved by removing or rearranging the Introduction-like sentences which make is sound too generic (for instance – “While wastewater is considered a burden on the shoulders of the countries as it consumes a large amount of energy before safe discharge, it can be an opportunity through proper treatment methods.”)

In general, The English language in this sections needs revision both in terms of sentences and expressions used (Example – “To prove the superiority of the proposed strategy, a comparison with previous data has been done.”. In my opinion replacing the words “prove” and “superiority” will contribute to the scientific soundness of the text.

Introduction section also needs English revision. As an example – first sentence sound odd and must be rearranged (“The high population growth and extremely energy consumption are the main challenges facing the world.”). Considering the above, I recommend English revision by a native speaker or someone with academic level language skills.

The materials and methods used are not described at the extend expected from a research paper. For instance: what is the external voltage applied to the MEC during the experiments; What is the internal resistance of the cell and how it changes in time; how the collected gas was analysed for hydrogen content? In single cell MEC reactor the collected gas could consist of carbon dioxide and other gases emitted during the anaerobic metabolism of bacteria instead of electrochemical produced hydrogen (produced by cathode reduction, not bi-hydrogen as stated by the authors). This issue need clarification since it could make all the excellent modelling and conclusions presented senseless.

Taking all these into account, at this stage I cannot recommend publication of the manuscript before major revision of the content.

Author Response

Dear reviewer

Thanks for your time and effort. We did our best to cover your comments.

The Abstract could be improved by removing or rearranging the Introduction-like sentences which make is sound too generic (for instance – “While wastewater is considered a burden on the shoulders of the countries as it consumes a large amount of energy before safe discharge, it can be an opportunity through proper treatment methods.”)

Response: Thanks for your suggestion. We updated the abstract to cover your comment.

In general, The English language in this sections needs revision both in terms of sentences and expressions used (Example – “To prove the superiority of the proposed strategy, a comparison with previous data has been done.”. In my opinion replacing the words “prove” and “superiority” will contribute to the scientific soundness of the text.

Response: Thanks for your suggestion. We replaced the words “prove” and “superiority. Also, We revised the paper accordingly.

Introduction section also needs English revision. As an example – first sentence sound odd and must be rearranged (“The high population growth and extremely energy consumption are the main challenges facing the world.”). Considering the above, I recommend English revision by a native speaker or someone with academic level language skills.

Response: Thanks for your comment. We revised the paper carefully  

The materials and methods used are not described at the extend expected from a research paper.

For instance: what is the external voltage applied to the MEC during the experiments.

 Response: Thanks for the reviewer’s comment. The applied voltage is 1.1 V.

 What is the internal resistance of the cell and how it changes in time;

 Response: Thanks for the reviewer’s comment.  In this study we focused on the hydrogen production in the MEC using an applied voltage of 1.1 V that was controlled using an external resistance of  10 W. The current values were measured using the voltage values across the external resistance (Rex).

how the collected gas was analysed for hydrogen content?

Response: Thanks for the reviewer’s comment. GC was used for the gas analysis.

In single cell MEC reactor the collected gas could consist of carbon dioxide and other gases emitted during the anaerobic metabolism of bacteria instead of electrochemical produced hydrogen (produced by cathode reduction, not bi-hydrogen as stated by the authors). This issue need clarification since it could make all the excellent modelling and conclusions presented senseless.

Response: Thanks for the reveiwer’s comment. We agree with the opinion of the reviewer, the collected gas would contain other gases such as CO2, N2, .etc. However, the hydrogen content of the gas was analysed using GC so we only calculate the bio-hydrogen. Furthermore, after each MEC batch cycle, the MEC reactor was undergone air-dried and exposed to atmospheric air for 30-45 min to inhibit the CH4 production process by suppressing methanogens.

Taking all these into account, at this stage I cannot recommend publication of the manuscript before major revision of the content

Response: Thanks for your suggestions. We did our best to cove all comments. 

Author Response

Response to Reviewer 1:

The manuscript submitted by Fathy et al. describes the Maximizing Bio-Hydrogen Production from an Innovative Microbial Electrolysis Cell Using Artificial Intelligence. The manuscript is presented well, the introduction and the results are written in good way. However, authors should carefully revise the manuscript based on the following comments before it can be considered further for publication.

Response: Thanks for your time and effort. We did our bet to cover your comments.

1- In the introduction session, the novelty and significance of the work should be emphasised. In addition, the potential impact of the research and why it is important, compared to other research in this field or previous studies, should be discussed.

Response: Thanks for your comment. We highlighted the novelty and significance of the work

2- There are several abbreviations and it would be better to make a list rather than to run through the text.

Response: Thanks for your comment. list of abbreviations is added and also We added the full name of acronyms

3- Please pay attention to some typos.

Response: Thanks for your comment. We revised the paper carefully

4- Did the authors evaluate the effect of concentration?

Response: Thanks for your comment. Fig. 4 illustrates the three- dimensional structure of the ANFIS model. The highest values of the output go to the dark red, but the lowest values go to the dark blue. The figure shows that the interaction between every two factors positively affects biohydrogen productivity compared to the single factor. As being clear from the figure that the increase of the initial COD resulted in increasing the biohydrogen production up to 60 %, then the biohydrogen production is decreased with the further increase in the COD concentration. The improved biohydrogen production by increasing the COD up to 60 % would be related to the increase in the COD (fuel) used by the microorganism for bio-hydrogen production, while at higher COD values, the biohydrogen decreased due to the poisoning effect of higher COD values on the microorganisms.

5- please cite the following ref (Magnetic Metal Oxide-Based Photocatalysts with Integrated Silver for Water Treatment) (Magnetic TiO2/CoFe2O4 Photocatalysts for Degradation of Organic Dyes and Pharmaceuticals without Oxidants)

Response: Thanks for your suggestion. We added these papers to the list of references

Reviewer 3 Report

In this manuscript, Ahmed Fathy et al. used a combination of neutron network model ANFIS and the Jellyfish Search Optimizer to optimize the conditions for microbial electrolysis cells. They found good match between experimental test data and the predicted values within a small database (16 points). The work overall could be informative, most of the conclusions are supported by presented data. I have a few questions.

1. Please clearly state the full name of acronyms when they first show up. For example, they never define COD. There are more examples, please double check.

2. Experiment. Did the authors confirm that the identity of the produced gas from the MEC? They seems to indicate that they only measured the volume of the produced gas using an inverted graduate cylinder.

3. The author may want to cite the following paper focusing on jellyfish search optimizer.

Jui-Sheng Chou, Dinh-Nhat Truong, Applied Mathematics and Computation 389 (2020) 125535.

4. Jellyfish optimizer. What are the values of the movement factor γ and Tmax in equations 7 and 9?

5. Page 8. The authors wrote “The optimal solutions are 30.0 oC, 7.3 and 60 (%) respectively for incubation temperature, initial pH and COD concentration.” From figure 6d, it seems like the best initial pH converges on pH 6.6. Why did the authors claim it is pH 7.3?

6. Table 3. The authors claimed that using the model-predicted conditions, they achieved 1.2664 m3 H2/m3d. Is this experimentally measured value or just predicted value? If it’s the latter, they authors need to explain this aspect in the manuscript.

Author Response

Response to Reviewer 2:

In this manuscript, Ahmed Fathy et al. used a combination of neutron network model ANFIS and the Jellyfish Search Optimizer to optimize the conditions for microbial electrolysis cells. They found good match between experimental test data and the predicted values within a small database (16 points). The work overall could be informative, most of the conclusions are supported by presented data. I have a few questions.

Response: Thanks for your time and effort. We did our bet to cover your comments.

  1. Please clearly state the full name of acronyms when they first show up. For example, they never define COD. There are more examples, please double check.

Response: Thanks for your comment. We added the full name of acronyms

  1. Experiment. Did the authors confirm that the identity of the produced gas from the MEC? They seems to indicate that they only measured the volume of the produced gas using an inverted graduate cylinder.

Response: Thanks for the reviewer’s comment. Yes, the volume of the gas was calculated by the displacement volume using the inverted cylinder. However, the gas analysis was performed using a gas chromatograph. This part was updated in the revised manuscript in the experimental part.

The following part was added:

The composition of the produced gas (H2, CH4, and N2) was analysed using GC “gas chromatography,  GC, model SRI 8600C, SRI Instruments, USA” that is armed with a He “helium” ionization and a TCD “thermal conductivity detector”.

  1. The author may want to cite the following paper focusing on jellyfish search optimizer.

Jui-Sheng Chou, Dinh-Nhat Truong, Applied Mathematics and Computation 389 (2020) 125535.

Response: Thanks for your suggestion. We added this paper to the list of references  

  1. Jellyfish optimizer. What are the values of the movement factor γ and Tmax in equations 7 and 9?

Response: Thanks for your comment. We mentioned the values in page 12 in the revised version.

  1. Page 8. The authors wrote “The optimal solutions are 30.0 oC, 7.3 and 60 (%) respectively for incubation temperature, initial pH and COD concentration.” From figure 6d, it seems like the best initial pH converges on pH 6.6. Why did the authors claim it is pH 7.3?

Response: Thanks for your comment. The corrected figures are presented in the revised version.

  1. Table 3. The authors claimed that using the model-predicted conditions, they achieved 1.2664 m3 H2/m3d. Is this experimentally measured value or just predicted value? If it’s the latter, they authors need to explain this aspect in the manuscript.

Response: Thanks for your comment. the proposed strategy contains two phases. The first phase is ANFIS modelling based on experimental data. Table 2 gives the verification results compared with the measured samples. Experimentally, the hydrogen production is 1.1747 m3 H2/m3 d under the following values 30.23 oC, 6.63 and 50.71 % respectively for incubation temperature, initial pH and COD concentration. Considering Table 2, the difference between measured value and the predicted by ANFIS is 0.0018 m3H2/m3d. It is 0.15 % from the reference value so it is considered a very small error and the results of ANFIS model is trustable. Then we used Jellyfish optimizer to determine the optimal temperature, initial potential of hydrogen, and influent COD concentration values.

Round 2

Reviewer 1 Report

The manuscript still needs English language revision (not only in grammar and style level)

For instance – the Introduction:

„Many approaches such as gasification, anaerobic digestion, and incineration are proposed to convert the wastes to chemical energy, energy carriers, such as ethanol, methanol, and hydrogen [8-13]. Among different energy carriers, hydrogen is the cleanest and most promising one due to the absence of venomous gases. Only water is produced, has a high energy density, can be used in many applications such as ammonia or methanol production and can be used as a transportation fuel. “

Here talking about conversion of the wastes to energy not all the mentioned forms are “chemical energy” as stated, namely incineration produce heat. “venomous gases” could be replaced with “potentially harmful gases”. There are many examples like this in the text. In my opinion, proofreading and revision by English native speaker could significantly improve the quality of the text.

Author Response

The manuscript still needs English language revision (not only in grammar and style level)

Response: Thanks for your time and effort. We revised the manuscript carefully.

For instance – the Introduction:

„Many approaches such as gasification, anaerobic digestion, and incineration are proposed to convert the wastes to chemical energy, energy carriers, such as ethanol, methanol, and hydrogen [8-13]. Among different energy carriers, hydrogen is the cleanest and most promising one due to the absence of venomous gases. Only water is produced, has a high energy density, can be used in many applications such as ammonia or methanol production and can be used as a transportation fuel. “

Here talking about conversion of the wastes to energy not all the mentioned forms are “chemical energy” as stated, namely incineration produce heat. “venomous gases” could be replaced with “potentially harmful gases”. There are many examples like this in the text. In my opinion, proofreading and revision by English native speaker could significantly improve the quality of the text.

Response: Thanks for your comment and recommendation. We did our best to improve the manuscript.

Reviewer 2 Report

I would like to thank all co-authors for their efforts.

Reviewer 3 Report

The authors addressed most of my comments. The response to my last comment is not satisfactory. My question was: is the value in table 3 under optimal conditions, 1.252 m3/m3d, just predicted value from the JO/ANFIS model, or is it also experimentally verified value? If it is the latter, the authors need to indicate that this is only predicted value, not experimentally confirmed ones, and further work is needed to confirm it. I was not asking about table 2.

Author Response

The authors addressed most of my comments.

Response: Thanks for your time and effort.

The response to my last comment is not satisfactory. My question was: is the value in table 3 under optimal conditions, 1.252 m3/m3d, just predicted value from the JO/ANFIS model, or is it also experimentally verified value? If it is the latter, the authors need to indicate that this is only predicted value, not experimentally confirmed ones, and further work is needed to confirm it. I was not asking about table 2.

Response: Thanks for you suggestion. This value is obtained by JO/ANFIS model. We added in the future work that the optimized results will be experimentally verified.

Round 3

Reviewer 1 Report

I have not specific comments at this stage.

Reviewer 3 Report

The authors addressed my previous comment.